# Myeloid NGS Analyses of Paired Samples from Bone Marrow and Peripheral Blood Yield Concordant Results: A Prospective Cohort Analysis of the AGMT Study Group

**DOI:** 10.3390/cancers15082305

**Published:** 2023-04-14

**Authors:** Bettina Jansko-Gadermeir, Michael Leisch, Franz J. Gassner, Nadja Zaborsky, Thomas Dillinger, Sonja Hutter, Angela Risch, Thomas Melchardt, Alexander Egle, Manuel Drost, Julian Larcher-Senn, Richard Greil, Lisa Pleyer

**Affiliations:** 1Salzburg Cancer Research Institute (SCRI), Center for Clinical Cancer and Immunology Trials (CCCIT), 5020 Salzburg, Austria; 23rd Medical Department with Hematology, Medical Oncology, Hemostaseology, Rheumatology and Infectiology, Oncologic Center, Paracelsus Medical University, 5020 Salzburg, Austria; 3Cancer Cluster Salzburg (CCS), 5020 Salzburg, Austria; 4Laboratory of Immunological and Molecular Cancer Research (LIMCR), 5020 Salzburg, Austria; 5Laboratory for Molecular Cytology (MZL), 5020 Salzburg, Austria; 6Department of Biosciences and Medical Biology, Allergy-Cancer-BioNano Research Centre, University of Salzburg, 5020 Salzburg, Austria; 7Austrian Group for Medical Tumor Therapy (AGMT) Study Group, 1180 Vienna, Austria; 8Assign Data Management and Biostatistics GmbH, 6020 Innsbruck, Austria

**Keywords:** next generation sequencing (NGS), concordance, peripheral blood, bone marrow, diagnosis, prognosis, myelodysplastic syndromes/neoplasms (MDS), acute myeloid leukemia (AML), myeloid neoplasias

## Abstract

**Simple Summary:**

Myelodysplastic neoplasms and acute myeloid leukemias are often caused by gene mutations. Next generation sequencing (NGS) has become indispensable for mutational assessment and is widely used for disease classification, risk stratification, prognostication, and disease monitoring. In these diseases, the bone marrow blast percentage and hence bone marrow specimen remain pre-requisite for the above. Several groups, including ours, report that bone marrow evaluations, which can be painful and time-consuming, are only performed in ~50% of patients during follow-up outside of clinical trials, indicating a clinical need for surrogate samples. We therefore aimed to compare NGS results for paired bone marrow and peripheral blood samples. Our results clearly show, in a prospective setting, that sequential molecular analyses of peripheral blood specimens can be reliably used to molecularly classify and monitor myeloid neoplasms without loss of sensitivity or specificity, even in the absence of circulating blasts or in neutropenic patients. Hence, a bone marrow evaluation for the purpose of monitoring of mutations is not necessary.

**Abstract:**

**Background**: Next generation sequencing (NGS) has become indispensable for diagnosis, risk stratification, prognostication, and monitoring of response in patients with myeloid neoplasias. Guidelines require bone marrow evaluations for the above, which are often not performed outside of clinical trials, indicating a need for surrogate samples. **Methods**: Myeloid NGS analyses (40 genes and 29 fusion drivers) of 240 consecutive, non-selected, prospectively collected, paired bone marrow/peripheral blood samples were compared. **Findings**: Very strong correlation (r = 0.91, *p* < 0.0001), high concordance (99.6%), sensitivity (98.8%), specificity (99.9%), positive predictive value (99.8%), and negative predictive value (99.6%) between NGS analyses of paired samples was observed. A total of 9/1321 (0.68%) detected mutations were discordant, 8 of which had a variant allele frequency (VAF) ≤ 3.7%. VAFs between peripheral blood and bone marrow samples were very strongly correlated in the total cohort (r = 0.93, *p* = 0.0001) and in subgroups without circulating blasts (r = 0.92, *p* < 0.0001) or with neutropenia (r = 0.88, *p* < 0.0001). There was a weak correlation between the VAF of a detected mutation and the blast count in either the peripheral blood (r = 0.19) or the bone marrow (r = 0.11). **Interpretation**: Peripheral blood samples can be used to molecularly classify and monitor myeloid neoplasms via NGS without loss of sensitivity/specificity, even in the absence of circulating blasts or in neutropenic patients.

## 1. Introduction

Acute myeloid leukemia, myelodysplastic neoplasms, and chronic myelomonocytic leukemia are malignant clonal hematologic neoplasms arising within the bone marrow [1]. These diseases are caused by genetic alterations in bone marrow stem cells leading to expansion of malignant blasts and hematopoietic insufficiency resulting in peripheral blood cytopenia [1,2,3]. While acute myeloid leukemia, myelodysplastic neoplasms, and chronic myelomonocytic leukemia are different disease entities, they share similar clinical features (e.g., accumulation of blasts, cytopenia, or recurrent infections) and are often treated similarly [2,4,5,6]. Diagnosis of acute myeloid leukemia, myelodysplastic neoplasms, and chronic myelomonocytic leukemia requires morphologic examination of the peripheral blood and bone marrow to quantify the amount of blasts (and blast equivalents), other cells (i.e., monocytes), and morphologic features (i.e., cellular dysplasia, cytopenia, presence of ring sideroblasts) [1,3,7,8,9,10].

Genetic analysis of the bone marrow has been performed for several decades using conventional karyotyping [1,7,8]. With the advances and widespread availability of NGS techniques, large scale genomic testing has been introduced into daily clinical practice in the last five to ten years. The broad application of next generation sequencing (NGS) techniques in recent years has resulted in increasing knowledge about the mutational landscape of myeloid neoplasias [11,12,13,14,15,16,17,18,19,20]. This is highly reflected in the recently published 5th edition of the World Health Organization Classification of Haematolymphoid Tumours [7] as well as in the International Consensus Classification of Myeloid Neoplasms and Acute Leukemias [8]. Herein, diseases that were historically defined by morphology only are now classified mainly on a molecular basis. This allows for better inter-observer comparison and forms the basis for targeted and tailored treatment approaches. Thus, both classifications increasingly rely on defining diseases and disease subtypes based on recurrent genetic abnormalities with prognostic and in some cases therapeutic relevance.

Morphologic bone marrow examinations are mandatory for (i) accurate diagnosis and disease classification (e.g., [7,8,11]), (ii) risk stratification, which typically also includes the bone marrow blast percentage as well as conventional cytogenetics and/or fluorescence in situ hybridization from bone marrow specimen (e.g., [21,22,23]), and (iii) response assessment/disease monitoring during treatment (e.g., [22,23,24,25,26,27]). In addition, most large-scale reports on molecular data in these diseases were generated from bone marrow specimen only and hence cannot eo ipso be extrapolated to results obtained in the peripheral blood [e.g., [28,29,30,31,32]). All current response criteria for these diseases require a bone marrow evaluation for determining adequate therapeutic response [8,22,23,24,25,26]. Histologic examination has to be awaited, which can be time consuming. However, rapid identification of genetic abnormalities with prognostic and therapeutic implications is becoming increasingly important for the treatment of patients with myeloid neoplasias in general, and of acute myeloid leukemia in particular, since treatment decisions heavily rely on the genomic disease profile (i.e., integration of targeted agents to standard induction therapy). Despite that, bone marrow aspirations are not always feasible in clinical practice (i.e., dry tap or insufficient aspiration) and, after having established the initial diagnosis, are only performed in approximately 50% of treated patients with myelodysplastic neoplasms, chronic myelomonocytic leukemia, or acute myeloid leukemia during follow-up [33,34,35,36]. Hence, a real-world clinical need exists to be able to use an alternative or surrogate sample, to be able to obtain information on the mutational status of a patient’s disease. In this regard, research groups around the world have assessed whether the genetic information required for an integrated diagnosis and/or treatment monitoring may also be obtained from analysis of the peripheral blood of patients with various hematologic malignancies (Table 1 and Appendix A) [20,37,38,39,40,41,42,43,44,45,46,47,48].

This monocentric, prospective analysis expands on the work by others, by evaluating the concordance of peripheral blood and bone marrow mutational analyses in a larger cohort of non-selected, consecutive adult patients treated at our institute for whom a myeloid NGS analysis was requested by the treating physicians. The commercially available AmpliSeq myeloid panel from Illumina including 40 DNA target genes and 29 RNA drivers was used with a high median (interquartile range (IQR)) read depth of 6116 (3675–9866), as well as a high minimum read depth of 500. Our main aim was to ascertain whether myeloid NGS analyses from peripheral blood could safely be used as an alternative to bone marrow samples to identify gene mutations and guide treatment decisions.

## 2. Materials and Methods

### 2.1. Patients and Cohort

The data collection and cleaning period was from 18 December 2019 to 11 November 2022. Database lock (last patient in) was on 13 October 2022. From 18 December 2019 to 11 November 2022 paired bone marrow and peripheral blood samples were obtained from non-selected, consecutive patients included in the Austrian Myeloid Registry (HMA cohort) and patients who were diagnosed and/or treated at the 3rd Medical Department, Laboratory for Molecular Cytology, Salzburger Landeskliniken (SALK), Paracelsus Medical University (PMU) (SALK cohort) and who had given written informed consent to this study. Patients with either a myeloid neoplasm or patients who underwent bone marrow examination during workup of a suspected myeloid disease were included. The sole inclusion criteria were the availability of a paired bone marrow and peripheral blood sample and the routine request for a myeloid NGS analysis of both materials by the treating physicians.

The Austrian Myeloid Registry of the Austrian Group of Medical tumor Therapy (AGMT) (NCT04438889; ethics committee approval 415-E/2581/Sept-2020) is a multicenter database that includes patients with acute myeloid leukemia, myelodysplastic neoplasms, and chronic myelomonocytic leukemia. This registry adheres to published quality guidelines of the U.S. Department of Health and Human Services Agency for Healthcare Research and Quality. All patients alive at the time of data entry into the electronic case report form provided written informed consent. Further details about this registry have been published previously [49].

All patients not included in the registry provided written informed consent that the NGS analyses may be performed, and that the data may be used for scientific analyses and publications.

### 2.2. Mutational Analyses

DNA was isolated from non-selected consecutive paired peripheral blood samples and bone marrow aspirates using Maxwell^®^ RSC Blood DNA Kit (Promega, Madison, WI, USA). The DNA concentration was measured using the QuantiFluor^®^ dsDNA System from Promega. The RNA concentration was measured using the QuantiFluor^®^ RNA System from Promega. The library preparation was performed using AmpliSeq™ library plus for Illumina^®^ using the AmpliSeq™ myeloid panel from Illumina^®^ (San Diego, CA, USA) according to the protocol of the AmpliSeq™ for Illumina^®^ myeloid panel reference guide. This NGS panel covers 40 DNA target genes (23 hotspot regions and 17 full genes) and 29 RNA fusion driver genes as detailed in Appendix A. All NGS libraries were quantified using the QuantiFluor^®^ dsDNA System from Promega. Buccal swab DNA NGS analysis was performed in patients harboring mutations with an allele frequency between 45 and 55%, or 95 and 100% to distinguish between germline and somatic mutation. The targeted sequencing was performed on the Illumina MiSeqDx and NextSeq550. Depending on the instrument, the starting concentration was 20 pM for the MiSeqDx and 1.6 pM for the NextSeq550.

A separate FLT3-ITD mutation analysis was performed using the LeukoStrat^®^ FLT3 Mutation Assay 2.0 from Invivoscribe (San Diego, CA, USA). This kit is based on fragment analysis using the ABI3500 genetic analyzer and the Genemapper 6 for visualization. 

### 2.3. Bioinformatic Analyses

The datasheets for sequencing were written using the Illumina local run manager. For read alignment and variant calling the following software tools were used: Local Run Manager DNA Amplicon Analysis Module (3.24.1.8+), Burrow-Wheeler Aligner Maximal Exact Match (BWA-MEM) [50,51] Whole-Genome Aligner (0.7.9a-isis-1.0.2), Pisces Variant Caller (5.2.9.23), Illumina Annotation Engine (2.0.11-0- g7fb24a09), binary alignment map (BAM) Metrics (v.0.0.22), and Sequence Alignment Map (SAM) tools (0.1.19-isis-1.0.3) [52]. Reads were aligned to the human genome hg19. Variant annotation and filtering were performed using Illumina VariantStudio v3.0 setting sensitivity to 1% variant allele frequency (VAF). Single nucleotide polymorphisms (SNPs) were excluded using the Exome Aggregation Consortium (ExAc) Database. Variants present in >1% of the general population, as indicated in the ExAc database, were defined as SNPs and were not included in the final data analysis. All variants were checked for sequencing artefacts using the BAM files and the Integrative Genomics Viewer (IGV) version 2.3.97 [53] for visualization. In case of discordance between bone marrow and peripheral blood samples, the discordant mutations were manually reviewed using the IGV and the corresponding BAM file. The sensitivity of the targeted NGS approach was set at 1% variant allele frequency (VAF), as it is not possible to set a lower threshold in the Local Run Manager. However, when searched manually for the mutations they could be found at lower thresholds. For mutation annotation the Catalogue of Somatic Mutations in Cancer (COSMIC) database and the database of single nucleotide polymorphism (dbSNP) were used. Variants annotated as benign or likely benign were not included in this study.

### 2.4. Statistical Analyses

To assess the correlation between mean VAF of the bone marrow and the mean VAF of the peripheral blood the Spearman correlation was used. Results were further illustrated by simple regression lines. If the VAF in the peripheral blood and in the bone marrow did not exactly match, the regression line deviated from the bisecting line.

Results were reported as significant when *p* ≤ 0.05 and a very strong correlation was reported when 0.90 < |r| < 1, a strong correlation was reported when 0.70 < |r| < 0.89, 0.40 < |r| < 0.69 indicates a moderate correlation, a weak correlation was reported when 0.10 < |r| < 0.39, and 0.00 < |r| < 0.10 indicates a negligible correlation [54]. Sensitivity analyses were performed using Kendall’s tau and by performing stratifications of paired samples (i.e. analyzing only those drawn on the same day) and of mutations (i.e., excluding found aberrations with a VAF < 5%).

As suggested by Bland et al. in the *Lancet* [55], a graphical representation of the agreement between the two methods of clinical measurement, i.e., mean of BM_VAF_ and PB_VAF_ versus (vs) the difference of BM_VAF_ and PB_VAF_, was generated. Only if both values (BM_VAF_ and PB_VAF_) were available, the mean and difference were calculated. 

Assign Data Management and Biostatistics GmbH (Innsbruck, Austria) performed statistical analyses with SAS^®^ 9.3. Life and Medical Sciences Institute, University of Bonn performed statistical analyses including mixed-effect linear modelling with Python 3.8.12.

## 3. Results

### 3.1. Patient Characteristics

Data from 187 non-selected, consecutive patients (108 patients from the HMA cohort and 79 from the SALK cohort) with paired bone marrow and peripheral samples were prospectively analyzed. Of these, 30 patients had more than one paired bone marrow and peripheral samples, so that the total number of paired bone marrow and peripheral samples was 240 (Figure 1). Patients with serial sample pairs had a median of 2 (IQR: 2–3, min–max: 2–5) sample pairs during the observation period. In total, 216 of 240 (90.0%) sample pairs were drawn on the same day, 5 (2.1%), 10 (4.2%), 8 (3.3%), and 1 (0.4%) sample pairs were drawn 1, >1–4, >4–8, and >8 week(s) apart, respectively.

Patient characteristics on the day of the bone marrow assessment are shown in Table 2. The median age of the entire cohort was 70 years (IQR = 58.0–77.7); 100 (53.5%) were male. Forty-six patients (24.6%) had a diagnosis of acute myeloid leukemia, 43 (23.0%) had myelodysplastic syndromes, 15 (8.0%) had a myelodysplastic/myeloproliferative overlap syndrome (MDS/MPN), 33 (17.6%) had a myeloproliferative neoplasm (MPN), and 50 (26.7%) did not have a myeloid neoplasm (termed “others” in Table 2). Ninety-one (48.7%) patients had a white blood cell count < 4000 G/L and 133 (71.1%) had no blasts in the peripheral blood count on the day of bone marrow aspiration. Treatment-related disease was present in 17 (9.1%) of 187 patients and 19 (10.2%) patients had a complex or monosomal karyotype, respectively.

### 3.2. Mutational Analyses—Overview

A total of 1321 variants were detected in 480 samples including 22 fusion genes, 910 single nucleotide variants (SNV), 186 insertions, 191 deletions and 12 multiple nucleotide polymorphisms (MNP). Off all detected variants, 994 had a COSMIC and/or dbSNP entry and 327 were not described in either of these two databases.

Mutations were found in 38 different genes in the bone marrow and 37 different genes in the peripheral blood, respectively. No mutations were detected in ABL1 or CSFR3 in all samples analyzed. In 41 patients and 94 samples no pathogenic mutations were detected as shown in Figure 1. The mean (SD) [min–max] number of mutations found in the bone marrow was 2.8 (2.4) [0–10] and in the peripheral blood the mean number of mutations was 2.7 (2.4) [0–10].

**Table 2 cancers-15-02305-t002:** Patient characteristics on the day of bone marrow assessment.

	Patients with PB and BM Sample Pairs (n = 187)	Total no. of PB and BM Sample Pairs (n = 240)
Mean days between PB and BM samples (SD)	2.2 (10.5)	2.2 (10.5)
Median (IQR)	0.0 (0.0–0.0)	0.0 (0.0–0.0)
Min–max	0–118	0–118
Unknown, n (%)	0 (0.0)	0 (0.0)
WHO 2016 Classification: MDS, n (%)	43 (23.0)	63 (26.3)
MDS/MPN	15 (8.0)	16 (6.7)
AML	46 (24.6)	72 (30)
MPN	33 (17.6)	38 (15.8)
Others ^1^	50 (26.7)	51 (21.3)
Unknown	0 (0.0)	0 (0.0)
Mean age (SD), years	66.1 (14.7)	65.2 (14.8)
Median (IQR)	70.0 (58.0–77.7)	68.5 (57.5–76.3)
Min–max	18–90	18–90
Unknown	0 (0.0)	0 (0.0)
Sex: Female, n (%)	87 (46.5)	115 (47.9)
Male	100 (53.5)	125 (52.1)
Unknown	0 (0.0)	0 (0.0)
Treatment-related disease: No, n (%)	170 (90.9)	218 (90.8)
Yes	17 (9.1)	22 (9.2)
Unknown	0 (0.0)	0 (0.0)
Normal karyotype: No, n (%)	53 (28.3)	60 (25.0)
Yes	108 (57.8)	134 (55.8)
Unknown	26 (13.9)	46 (19.2)
Complex karyotype: No, n (%)	148 (79.1)	180 (75.0)
Yes	13 (6.9)	14 (5.8)
Unknown	26 (13.9)	46 (19.2)
Monosomal karyotype: No, n (%)	155 (82.9)	188 (78.3)
Yes	6 (3.2)	6 (2.5)
Unknown	26 (13.9)	46 (19.2)
Peripheral blood blasts, %: Mean (SD)	5.8 (16.5)	5.0 (14.9)
Median (IQR)	0.0 (0.0–2.0)	0.0 (0.0–1.0)
Min–max	0.0–99.0	0.0–99.0
Unknown, n (%)	0 (0.0)	3 (1.3)
Bone marrow blasts histology, %: Mean (SD)	9.0 (18.9)	10.1 (20.7)
Median (IQR)	2.5 (2.5–2.5)	2.5 (2.5–2.5)
Min–max	0.0–95.0	0.0–95.0
Unknown, n (%)	21 (11.2)	41 (17.2)
Bone marrow blasts aspirate, %: Mean (SD)	12.7 (25.1)	14.0 (25.9)
Median (IQR)	2.0 (1.0–6.0)	2.0 (1.0–8.0)
Min–max	0.0–100.0	0.0–100.0
Unknown, n (%)	31 (16.6)	43 (17.9)
White blood cell count, G/L: Mean (SD)	13.9 (34.6)	11.6 (30.9)
Median (IQR)	4.8 (2.7–9.1)	4.2 (2.2–8.7)
Min–max	0.6–305.6	0.5–305.6
Unknown, n (%)	0 (0.0)	0 (0.0)
Absolute neutrophil count, G/L: Mean (SD)	7.9 (23.9)	6.6 (21.3)
Median (IQR)	2.7 (1.2–5.3)	2.3 (0.8–4.8)
Min–max	0.0–226.1	0.0–226.1
Unknown, n (%)	0 (0.0)	0 (0.0)
Monocytes, %: Mean (SD)	9.2 (9.6)	9.6 (10.1)
Median (IQR)	6.8 (3.0–12.0)	7 (3.0–12.0)
Min–max	0.0–72.0	0.0–72.0
Unknown, n (%)	0 (0.0)	1 (0.4)
Lymphocytes, %: Mean (SD)	27.7 (20.1)	29.6 (21.18)
Median (IQR)	23.0 (13.0–36.0)	25.0 (13.8–40.0)
Min–max	0.9–95.0	0.9–98.0
Unknown, n (%)	0 (0.0)	1 (0.4)
Hemoglobin, g/dL: Mean (SD)	106 (2.5)	10.5 (2.4)
Median (IQR)	10.3 (8.8–12.3)	10.1 (8.8–12.2)
Min–max	5.8–17.3	5.6–17.3
Unknown, n (%)	0 (0,0)	0 (0.0)
Mean cell volume, fl: Mean (SD)	92.7 (9.0)	92.6 (9.1)
Median (IQR)	91.4 (86.7–97.6)	91.3 (86.3–97.6)
Min–max	62.6–120.1	62.6–120.1
Unknown, n (%)	1 (0.5)	1 (0.4)
Mean cell hemoglobin, pg: Mean (SD)	31.7 (3.53)	31.6 (3.5)
Median (IQR)	31.3 (29.6–33.7)	31.2 (29.5–33.6)
Min–max	18.7–44.4	18.7–44.4
Unknown, n (%)	1 (0.5)	1 (0.4)
Platelet count, G/L: Mean (SD)	198.5 (234.1)	191.3 (219.1)
Median (IQR)	131.0 (58.0–223.0)	132.0 (53.5–230.5)
Min–max	6–1893	6–1893
Unknown, n (%)	0 (0,0)	0 (0,0)
Ferritin, µg/L: Mean (SD)	784.3 (1003.5)	1030.7 (1607.6)
Median (IQR)	412.0 (189.5–1001.5)	467.5 (196.5–1382.0)
Min–max	16–7212	11.0–1346
Unknown, n (%)	79 (42.2)	116 (48.3)
Creatinine, mg/dL: Mean (SD)	1.1 (0.83)	1.0 (0.75)
Median (IQR)	0.9 (0.7–1.1)	0.9 (0.7–1.1)
Min–max	0.3–9.5	0.3–9.5
Unknown, n (%)	8 (4.3)	13 (5.4)
Bilirubin, mg/dL: Mean (SD)	9.8 (0.9)	0.7 (0.85)
Median (IQR)	0.5 (0.4–0.8)	0.5 (0.4–0.8)
Min–max	0.1–8.0	0.1–8.0
Unknown, n (%)	9 (4.8)	15 (6.3)

BM indicates bone marrow; PB, peripheral blood; WHO, World health organization; MDS, myelodysplastic neoplasia; MDS/MPN, myelodysplastic neoplasia/myeloproliferative neoplasia; AML, acute myeloid leukemia; MPN, myeloproliferative neoplasia; SD standard deviation; IQR, interquartile range. ^1^ Includes evaluation for unexplained cytopenia(s) and/or cytoses due to the following diagnoses: B-cell lymphomas (n = 10), myeloma (n = 7), solid tumors (n = 7), rheumatic diseases (n = 3), pernicious anemia (n = 3), reactive cytoses (n = 3), aplastic anemia (n = 2), drug-induced bone marrow toxicity (n = 2), autoimmune hemolytic anemia/kryoglobulinemia (n = 2), liver cirrhosis (n = 2), familial Mediterranean fever (n = 1), human immunodeficiency virus infection (n = 1), hyper-eosinophilic syndrome (n = 1), cutaneous mastocytosis (n = 1), and unexplained cytopenias (n = 5).

### 3.3. Mutational Analyses—Concordance and Predictive Value

In total, there were 702 concordant (i.e., 656 gene mutations found in both peripheral blood and bone marrow paired samples and 46 sample pairs in which no mutations were found in either the peripheral blood or the bone marrow) and 9 discordant (of which 8 were found only in the bone marrow and 1 only in the peripheral blood) reported results (Table 3). A total of 1840 presumed concordant negative calls (i.e., the number of negative sample pairs (n = 46) multiplied by the number of genes in the panel (n = 40)) were detected, resulting in an overall concordance of 99.6% between the peripheral blood and the bone marrow, with a sensitivity of 98.8% and a specificity 99.9%. The positive predictive value (PPV) was 99.8% and the negative predictive value (NPV) was 99.6% (Table 3).

### 3.4. Mutational Analyses—Occurrence of Mutations

There was strong correlation between the occurrence (i.e., the detection) of mutations in peripheral blood versus bone marrow samples (r = 0.91; *p* < 0.0001). The occurrence of mutations by gene and sample type for all samples of the total cohort are depicted in Figure 2. In both sample types, the five most commonly mutated genes were TET2 (25.4% vs. 25.4%), ASXL1 (24.6% vs. 24.6%), DNMT3A (21.3% vs. 21.3%), SRSF2 (12.5% vs. 12.1%), and RUNX1 (11.3% vs. 10.8%) for bone marrow vs. peripheral blood, respectively (Figure 2). Two patients harbored a FLT3-ITD mutation, which was detected in the bone marrow and the peripheral blood in all sample pairs (n = 2). All detected fusion genes (n = 11) were detected in both the bone marrow and the peripheral blood in all sample pairs (n = 11).

Below, the occurrence of mutations by gene and sample type (bone marrow vs. peripheral blood) is given separately for the five most commonly mutated genes for each diagnostic subgroup (further details in Appendix A):**Acute myeloid leukemia**: DNMT3A (31.9% vs. 31.9%), NPM1 (19.4% vs. 18.1%), IDH2 (19.4% vs. 19.4%), TET2 (19.4% vs. 19.4%), and TP53 (15.3% vs. 15.3%) (Appendix A).**Myelodysplastic neoplasms**: TET2 (37.1% vs. 37.1%), ASXL1 (33.9% vs. 33.9%), DNMT3A (21.0% vs. 21.0%), TP53 (17.7% vs. 17.7%), and RUNX1 (17.7% vs. 17.7%) (Appendix A).**Myelodysplastic/myeloproliferative overlap syndromes**: ASXL1 (52.9% vs. 52.9%), TET2 (41.2% vs. 41.2%), SRSF2 (35.3% vs. 35.3%), NRAS (29.4% vs. 29.4%), and RUNX1 (23.5% vs. 23.5%) (Appendix A).**Myeloproliferative neoplasms**: ASXL1 (34.2% vs. 34.2%), TET2 (31.6% vs. 31.6%), JAK2 (31.6% vs. 31.6%), SRSF (23.7% vs. 21.1%), and CALR (21.1% vs. 21.1%) (Appendix A).**Other (i.e., non-myeloid) diagnoses**: DNMT3A (15.7% vs. 15.7%), ASXL1 (13.7% vs. 13.7%), TET2 (9.8% vs. 9.8%), MYD88 (7.8% vs. 7.8%), and CBL (7.8% vs. 7.8%) (Appendix A).

**Figure 2 cancers-15-02305-f002:**
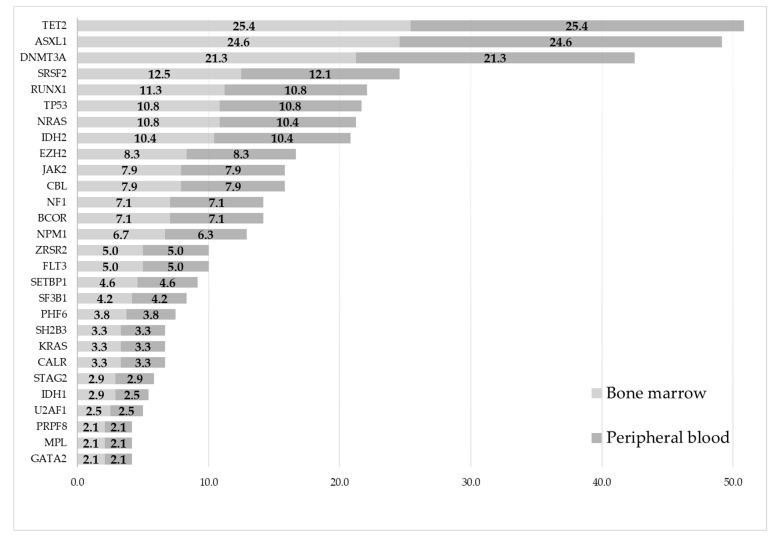
Occurrence of mutations by gene and by sample type for all samples (n = 150 patients, n = 191 sample pairs ^1,2^, n = 1299 mutations detected ^3,4^). The X-axis represents the percentage of samples in which the respective mutation was found. (^1^ Includes 53 serial sample pairs from 30 patients. ^2^ Total sample pairs (n = 240), excluding sample pairs without mutations (n = 46) and excluding sample pairs in which only fusion genes (and no other mutations) were found (n = 3). ^3^ Mutations occurring in genes in <2% of the total cohort were not included in the graph and included the following genes for bone marrow vs. peripheral blood, respectively: MYD88 1.7% vs. 1.7%; ETV6 1.7% vs. 1.7%; CEBPA 1.7% vs. 1.7%; PTPN11 1.3% vs. 1.3%; IKZF 1.3% vs. 1.3%; HRAS 1.3% vs. 1.3%; BRAF 0.8% vs. 0.8%; WT1 0.8% vs. 0.8%; KIT 0.8% vs. 0.8%; and RB1 0.4% vs. 0.0%. ^4^ If a sample had more than one mutation in the same gene, this was only counted one time).

### 3.5. Mutational Analyses—Correlation of BM_VAF_ vs. PB_VAF_

In 92 samples (i.e., in 46 sample pairs) no mutations were found in either the peripheral blood or the bone marrow. In all analyzed samples in which mutations were detected (excluding fusion genes and FLT3-ITD as the methods used do not produce VAF values, resulting in a remaining n = 382 samples, i.e., n = 191 sample pairs), the mean (SD) [min–max] of VAFs was 25.8% (22.9) [0.1–99.7%] in the bone marrow and 23.1% (21.8) [0.1–99.8%] in the peripheral blood, respectively. Appendix A lists the mean BM_VAF_ and the mean PB_VAF_ by gene, respectively. Appendix A gives further information by diagnosis.

The strength of the correlation between the BM_VAF_ and the PB_VAF_ was calculated using the Spearman correlation coefficient, which was r = 0.93 (*p* < 0.0001), thus indicating a very strong correlation (visualized in Figure 3). Sensitivity analyses were performed using Kendall’s tau and the results remained nearly identical.

To determine whether the correlation might be different for patients with a potentially reduced tumor load in the peripheral blood, i.e., in patients with 0% peripheral blood blasts or with <1.0 G/L absolute neutrophil count, we performed the respective analyses for these patient subgroups (Figure 4). The correlation remained very strong for patients in all subgroups, i.e., in patient samples with either peripheral blood blasts = 0% (n = 165; r = 0.92, *p* < 0.0001) or ≥1% (n = 74, r = 0.95, *p* < 0.0001), and with an absolute neutrophil count of <1.0 G/L (n = 66; r = 0.88, *p* < 0.0001) or ≥1.0 G/L (n = 174; r = 0.95, *p* < 0.0001).

To determine whether the bone marrow blast percentage might correlate with the BM_VAF_ and/or whether the peripheral blood blast percentage correlated with the PB_VAF_, we plotted these values against each other in Figure 5A,B. The Spearman correlation coefficients were low, with r = 0.11 (*p* < 0.004) in the bone marrow (Figure 5A) and r = 0.19 (*p* < 0.0001) in the peripheral blood (Figure 5B), indicating a weak correlation between blast counts and the detected VAFs.

**Figure 4 cancers-15-02305-f004:**
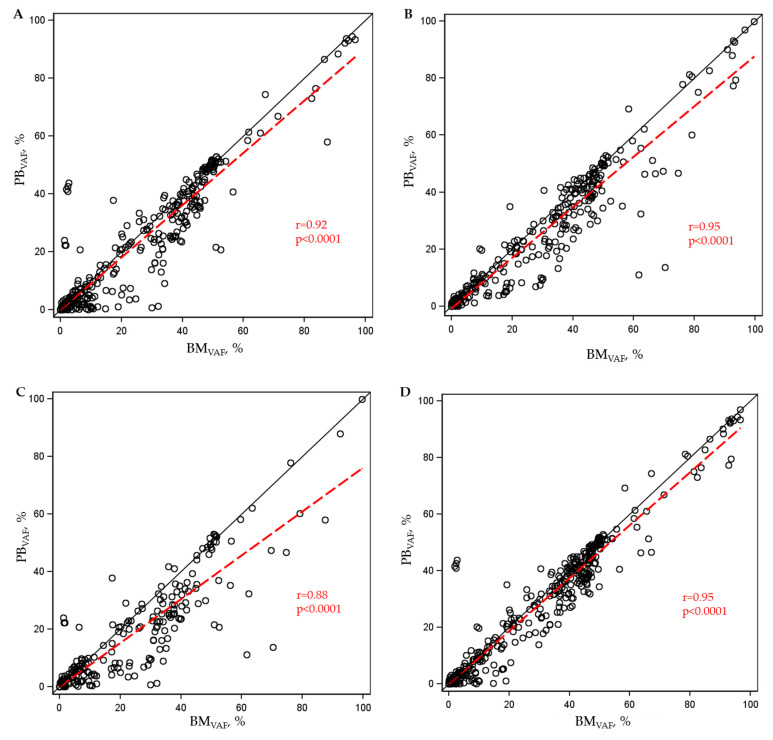
Scatterplot of the variant allele frequency (VAF) of the mutated genes in bone marrow and peripheral blood for patient samples with the following peripheral blood parameters on the day of bone marrow sampling. (**A**) Peripheral blast percentage = 0% (n = 165). (**B**) Peripheral blast percentage ≥ 1% (n = 74). ^1^ (**C**) Absolute neutrophil count < 1.0 G/L (n = 66). (**D**) Absolute neutrophil count ≥ 1.0 G/L (n = 174). The black line is the bisecting line showing a perfect linear regression with slope = 1 and intercept = 0. The red dashed line is the regression line, indicating the correlation between BM_VAF_ and PB_VAF_ of each paired sample. (^1^ The peripheral blood blast percentage was missing for one patient).

### 3.6. Mutational Analyses—Agreement

A graphical representation of the agreement between the two methods of clinical measurement, i.e., BM_VAF_ versus (vs.) PB_VAF_, was generated according to Bland et al. [55] (Figure 6). The x-axis shows the mean of the two values, while the y-axis displays the difference between the two. The central red dashed line represents the mean of the difference ($\overset{-}{d}$ = 2.92), and the outer red dashed lines correspond to $\overset{-}{d}\pm 2\overset{-}{\sigma }$, where $\overset{-}{\sigma }$ represents the standard deviation of the difference ($\overset{-}{\sigma }$ = 7.71). As discussed in [55], the region between these two outer lines is referred to as the “limits of agreement” and it visualizes the difference between the two methods of measurement.

The results presented in Figure 6 indicate that both methods of clinical measurement (i.e., NGS from peripheral blood vs. NGS from bone marrow) are in good agreement with each other, as the mean of the difference is close to zero and the majority of the differences between the two methods fall within the “limits of agreement” (i.e., between the two central dashed red lines). This suggests that both sample types produce similar results and provide consistent measures of the clinical parameters under study.

Data points lying outside of the “limits of agreement” are termed outliers (n = 35). These occurred in the following genes: DNMT3A (n = 8), IDH2 (n = 6), RUNX1 (n = 6), TP53 (n = 4), NPM1 (n = 3), TET2 (n = 2), PHF6 (n = 2), and BCOR, EZH2, IDH1, JAK2 and NRAS (n = 1 each).

### 3.7. Discordant Mutations

Of 1321 detected mutations, 9 (0.68%) were found in only one of the paired samples, i.e., were discordant (Table 4). Discordant mutations included six SNVs, two insertions and one deletion (Table 4). Of these, only one mutation (in gene ASXL1, VAF 1.0%, read depth 5014) was found exclusively in the peripheral blood but not in the bone marrow. Only eight mutations were found exclusively in the bone marrow but not in the peripheral blood, namely NPM1 (VAF 0.6%, read depth 1724), SETBP1 (VAF 1.1%, read depth 6565), NRAS (VAF 1.3%, read depth 18517), RB1 (VAF 1.9, read depth 12772), SRSF2 (VAF 2.2%, read depth 960), ASXL1 (VAF 2.5%, read depth 5816), RUNX1 (VAF 3.7%, read depth 4955), and IDH1 (VAF 9.1%, read depth 4150). The mean (SD) [min–max] VAF of all mutations exclusively found in the bone marrow was 2.7% (2.7) [0.6–3.6%]. All mutations, except the IDH1 mutation in a patient with Waldenstrom’s disease, had a VAF of <4% (Table 4). Of these patients, four had a serial sample analysis. Further details are shown in Appendix A.

### 3.8. Further Subanalyses

The following further subanalyses were performed: (i) limiting the analyses to those patients for whom the paired samples were drawn on the same day, (ii) exclusion of all found aberrations with a VAF < 5%, and (iii) using the filters applied in both (i) and (ii). All results were similar to those reported above (Appendix A).

## 4. Discussion

In this prospective, monocentric analysis we report on the concordance of NGS results between paired samples obtained from the peripheral blood and the bone marrow samples of patients for whom a myeloid NGS panel was requested from our laboratory to inform their diagnostic and therapeutic decisions. 

To date, bone marrow biopsies are still required to correctly classify diseases [7,8] as well as to determine (molecular) response [8,22,23,24,25,26]. As mentioned above, however, bone marrow evaluations are only performed in approximately 50% of patients with myeloid neoplasias during follow-up [33,34,35,56]. Reasons why bone marrow evaluations are often not performed outside of clinical trials include the fact that response can often be determined from clinical benefit (e.g., loss of transfusion dependence, normalization of blood counts, disappearance of peripheral blood blasts, improvement of quality of life, and performance status). Similarly, disease progression can be inferred from clinical deterioration. Other reasons include a lack of clinical consequences (e.g., due to lack of alternative treatment options), anticipated bleeding complications, and logistic reasons. The patients may also decline repeated bone marrow evaluations [35]. Thus, a real-world clinical need exists to be able to obtain information on the mutational status of a patient’s disease without the requirement for repetitive bone marrow evaluations, especially as frequent molecular monitoring is desirable for adequate risk stratification [31,57], prognostication [15,32], and for adequate clinical management [58,59,60,61].

In this regard, evidence from retrospective analyses and clinical trials is accumulating that bone marrow blast clearance [22,23,24,25,26] and even bone marrow assessments may not be mandatory to determine whether meaningful clinical response with overall survival prolongation is achieved [34,35,56]. Furthermore, high complete concordance (63–100%) of genomic, cytogenetic, and phenotypic aberrations between paired peripheral blood and bone marrow samples has been found in patients with myeloid neoplasias using methods such as fluorescence in situ hybridization, chromosome banding arrays, or flow cytometry [37,38,39,46,47,48] (Appendix A). Peripheral blood samples have been identified as a viable alternative to bone marrow samples for monitoring cytogenetic data using fluorescence in situ hybridization [37,38,39,48] and flow cytometry [47] panels, as well as by conventional karyotyping [37] and SNP arrays [46].

Some research groups used targeted NGS approaches to assess whether peripheral blood may be an acceptable alternative sample to bone marrow in patients with hematologic neoplasms [40,41,42,43] (Table 1). These studies reported on 16–183 paired peripheral blood and bone marrow samples and detected a complete concordance of 69–97%. Four of these seven studies either included or were exclusively performed in patients with lymphoid neoplasms [40,42,44,45], five did not report how many days were allowed between the peripheral blood and bone marrow samples, two reported up to 334 days between the sampling, and the sensitivity of the NGS analyses and/or the mean (min–max) coverage were rarely reported. Nevertheless, these important studies underscore the relevance and the real-world clinical need to be able to diagnose and monitor treatment response without repeated bone marrow evaluations.

Our data show an exceptionally high concordance (99.6%), sensitivity (98.8%), specificity (99.9%), positive predictive value (99.8%), and negative predictive value (99.4%) between reported results obtained by NGS analyses of paired peripheral blood and bone marrow specimens.

To our best knowledge these results are the first to show a very strong correlation not only between the mutations detected in the peripheral blood and the bone marrow (r = 0.91; *p* < 0.0001), but also a very strong correlation between the VAFs detected in the peripheral blood and the bone marrow (r = 0.93; *p* < 0.0001). 

We are also the first to analyze and show that the concordance of the VAFs of detected mutations between the bone marrow and the peripheral blood was similarly high in patient subgroups with no circulating blasts (r = 0.92; *p* < 0.0001) or with neutropenia (as defined by an absolute neutrophil count of < 1.0 G/L) (r = 0.88; *p* < 0.0001). These results are in line with the fact that clonal involvement has been shown for maturing myeloid cells such as neutrophils [62] as well as for all accessory cells (except T-cells), including natural killer cells [63,64,65,66,67,68], myeloid/lymphoid dendritic cells [36,67,69], monocytes [67], and B-cells [66,70,71,72,73], all of which have been shown to bear the same cytogenetic abnormalities as the malignant bone marrow myeloid progenitors and blasts in patients with myelodysplastic neoplasms or acute myeloid leukemia.

These data are also the first to show a lack of correlation between the blast percentage in the bone marrow and BM_VAF_ of mutations, and the same accounted for the peripheral blood blast percentage and PB_VAF_ of mutations. This is an important finding for daily clinical practice, since clinicians can safely rely on the results of peripheral blood NGS analysis, when a bone marrow evaluation is not evaluable or not feasible, even in the absence of circulating blasts or in neutropenic patients.

Of 1321 found mutations, only 9 (0.68%) were discordant between bone marrow and peripheral blood. Eight of the 9 discordant mutations were found only in the bone marrow and one was found only in the peripheral blood. Most discrepancies (eight of nine mutations that were found in only one sample) were caused by low-level sub-clonal events (VAF ≤ 3.7%). The clinical relevance of VAFs < 5% is currently thought to be low [74] and most laboratories specify a sensitivity of their NGS methods of ≥5%. A VAF of ≥2% is required for the diagnosis of clonal hematopoiesis of indeterminate potential [75] and a VAF of ≥10% is required for recent classification systems [7,22].

Our results thus confirm and expand on reports by others, in an extremely well documented patient population with prospectively collected paired peripheral blood and bone marrow samples, >90% of which were drawn on the same day. Our group used the highest read depth of those reported and also had the highest concordance, indicating that increasing the read depth (and thus sensitivity) of the method increases concordance. Hence, it will be extremely interesting and relevant to perform similar studies for minimal residual disease analyses by NGS, once such tests become commercially available.

These findings support current guidelines, which advise against a repeated NGS analysis of bone marrow samples if a mutation has already been found in the peripheral blood [20].

## 5. Conclusions

Taken together, our results clearly show, in a prospective setting, that sequential molecular analyses of peripheral blood specimens can be reliably used to molecularly classify and monitor myeloid neoplasms without loss of sensitivity or specificity, and that a bone marrow evaluation solely for the purpose of monitoring of mutations is not necessary in (almost) all cases. These data thus ascertain that myeloid NGS analyses from peripheral blood can safely be used as an alternative to bone marrow samples to identify and monitor gene mutations and to guide treatment decisions. This allows for less frequent follow-up of bone marrow evaluations, which may perhaps be entirely omitted in the future. This is extremely relevant information for both the treating physicians and patients, as samples of peripheral blood can be drawn easily, nearly painlessly, and at multiple time points. NGS from peripheral blood specimen is thus of particular clinical interest for its use as a minimally invasive screening tool for diagnostic and therapy monitoring, especially in special situations such as a fibrotic or hypocellular marrow.

## Figures and Tables

**Figure 1 cancers-15-02305-f001:**
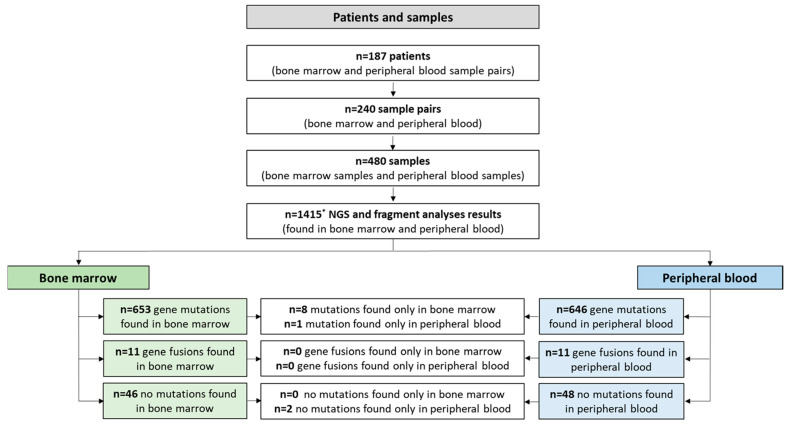
Numbers of patients, samples and mutations. * Total reported molecular results (n = 1415) consist of: 1321 pathogenic variants (8 fragment analysis, 1313 NGS); 94 samples with no mutations.

**Figure 3 cancers-15-02305-f003:**
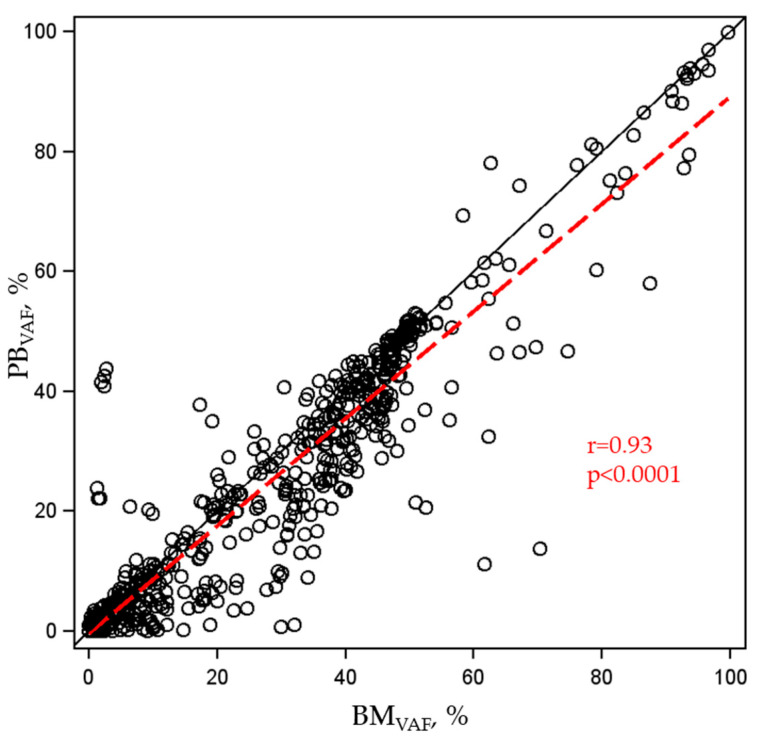
Scatterplot of the variant allele frequency (VAF) of the mutated genes in bone marrow and peripheral blood. The black line is the bisecting line showing a perfect linear regression with slope = 1 and intercept = 0. The red dashed line is the regression line, indicating the correlation between BM_VAF_ and PB_VAF_ of each paired sample.

**Figure 5 cancers-15-02305-f005:**
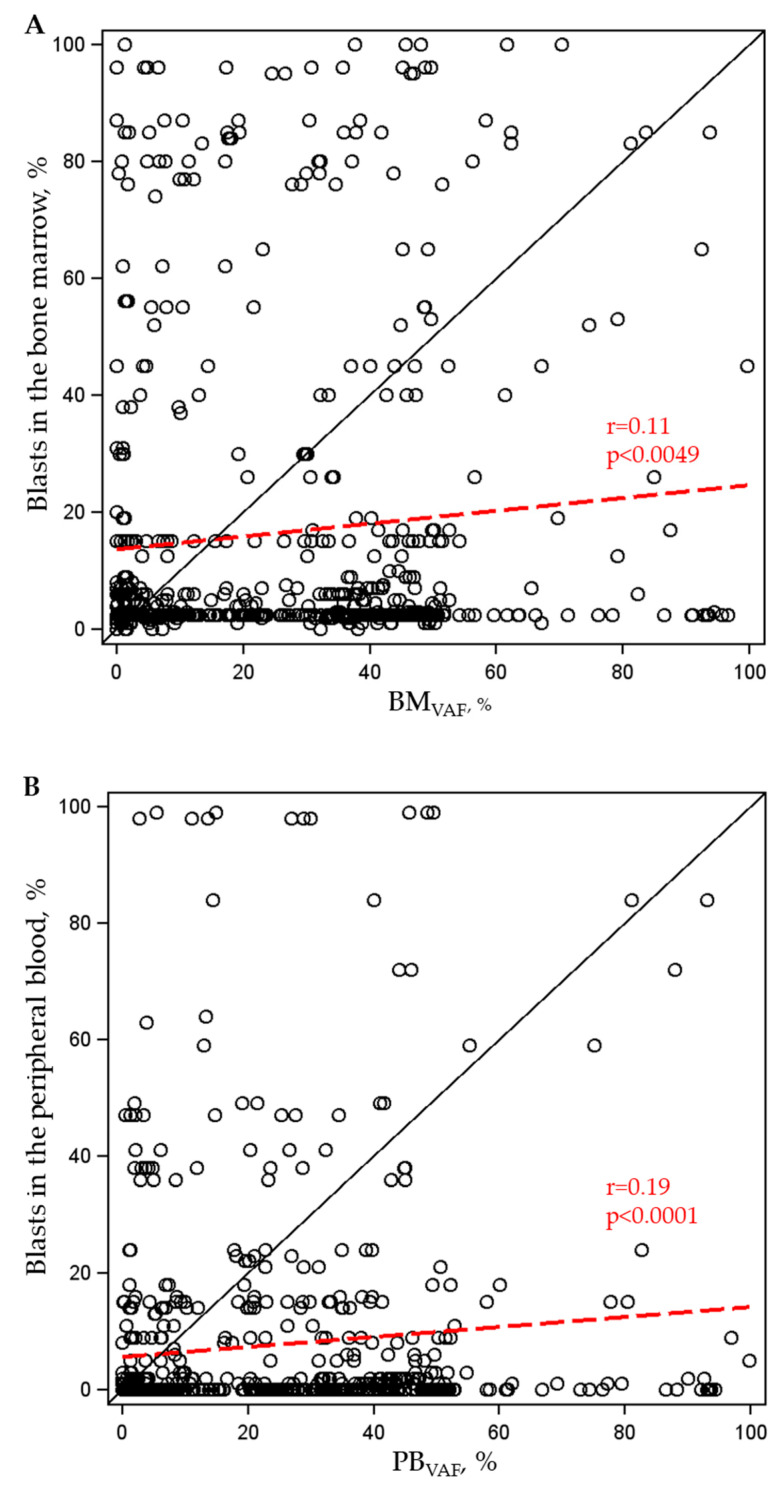
Scatterplots of the “variant allele frequency (VAF) of a mutation” vs. “blast percentage”. (**A**) “BM_VAF_” vs. “percentage of bone marrow blasts”. (**B**) “PB_VAF_” vs. “percentage of peripheral blood blasts”. The black line is the bisecting line showing a perfect linear regression with slope = 1 and intercept = 0. The red dashed line is the regression line, indicating the correlation between BM_VAF_ and PB_VAF_ of each paired sample.

**Figure 6 cancers-15-02305-f006:**
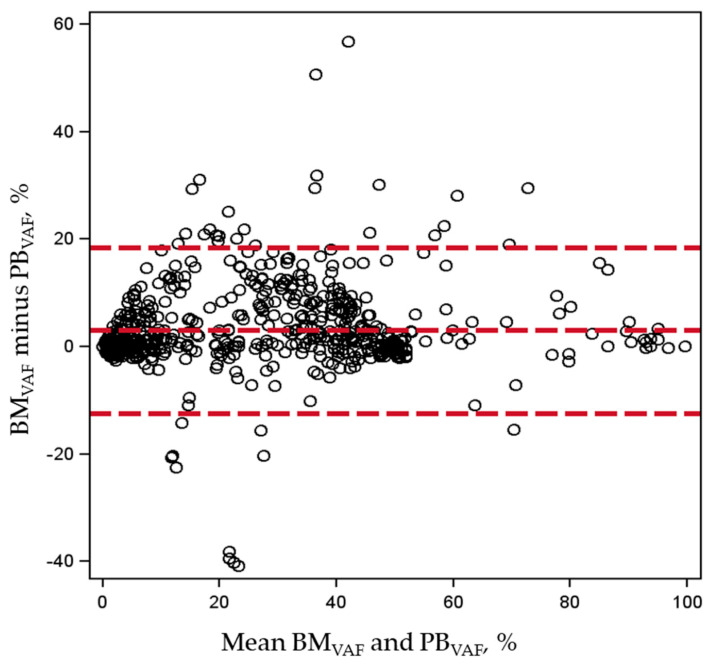
Scatterplot of the variant allele frequency (VAF): difference against the mean. ^1^ The X−axis shows the mean of the BM_VAF_ and the PB_VAF_. The Y−axis displays the difference between the BM_VAF_ and the PB_VAF_. The central red dashed line represents the mean of the difference, ($\overset{-}{d}$ = 2.92), and the outer red dashed lines correspond to $\overset{-}{d}\pm 2\overset{-}{\sigma }$, where $\overset{-}{\sigma }$ represents the standard deviation of the difference ($\overset{-}{\sigma }$ = 7.71). As discussed in [55], the region between these two outer lines is referred to as the “limits of agreement” and it visualizes the difference between the two methods of measurement. (^1^ Analyzed according to Bland et al., *Lancet* 1986 [55]).

**Table 1 cancers-15-02305-t001:** Published articles comparing NGS analyses from bone marrow and peripheral blood in patients with MDS, CMML, and AML ^1^.

First Author	Pts, n	Paired Samples, n	Disease, n/n (%)	Method	Company, kit	Days between BM-PB Analyses, Mean (min–max)	Concordance between Paired Samples (BM and PB)	Concordance between Paired Mutations (BM and PB)	Coefficient
Jansko-Gadermeir B. [current manuscript]	187	240	AML, 46/187 (24.6%) MDS, 43/187 (23.0%) MDS/MPN, 15/187 (8.0%) MPN, 33/187 (17.6%) Others, 50/187 (26.7%)	NGS	Illumina® AmpliSeq™ myeloid panel (40 genes, 29 driver fusion genes) Leukostrat Invivoscribe 2.0 (FLT3-ITD/TKD)	2 (0–118)	Complete concordance: 231/240 (96%) Partial concordance: 7/240 (3%)	Concordance: 702/711 (99.7%)	r = 0.93 *p* < 0.0001
Jumniensuk C. [42]	163	163	Cytopenia, 54/163 (33%) NHL, 31/163 (19%) AML, 23/163 (14%) MDS, 53/163 (13%) MPN, 21/163 (13%) MDS/MPN, 11/163 (7%) Others, 2/163 (1%)	NGS	Illumina® TruSight (54 genes)	63 (0–334)	Complete concordance: 124/163 (76%) Partial concordance: 26/163 (16%)	Concordance: not given	κ = 0·79 *p* < 0·0001
Stasik S. [43]	29	35	MDS, 2/40 (5%) AML, 38/40 (95%)	NGS (CD34^+^ MRD)	Life Technologies custom panel (4 genes)	Not reported	Complete concordance: not given Partial concordance: not given	Concordance: not given	r = 0·90 *p* <0·0001
Muffly L. [44]	62	126	T-ALL, 8/62 (13%) B-ALL, 54/62 (87%)	NGS (MRD)	Adaptive Biotechnologies clonoSEQ assay (TCR rearrangement)	Not reported	Complete concordance: 112/126 (89%) Partial concordance: not applicable	Concordance: not given	r = 0·87 *p* <0·0001
Ruan M. [41]	20	20	Pediatric AML, 20/20 (100%)	NGS	AcornMed Biotechnology customized Gene Panel (137 genes)	Not reported	Complete concordance: 155/209 (74%) Partial concordance: not given	Concordance: 155/239 (74%)	r = 0·95 *p* < 0·001
Lucas F. [40]	164	164	Myeloid neoplasias, 129/164 (79%) Lymphoid neoplasm, 32/164 (20%) MPAL, 3/126 (1.8%)	NGS	Rapid Heme Panel (95 genes)	2 (0–14)	Complete concordance: 130/164 (79%) Partial concordance: not given	Concordance: 278/329 (84.5%)	Not given
Fries C. [45]	16	16	B-ALL, 16/16 (100%)	NGS	IGH Vh-DJh rearrangement	Not reported	Complete concordance: 11/16 (69%) Partial concordance: 4/16 (25%)	Concordance: 23/28 (82.1%) ^2^	Not given
Mohmedali A.M. [46]	183	183	MDS, 183/183 (100%)	NGS	Illumina custom panel (24 genes)	Not reported	Complete concordance: 177/183 (97%) Partial concordance: not given	Concordance: 234/240 (97.5%)	Not given

^1^ Search terms used in PubMed from 5 June 2022 until the 7 December 2022 were: Concordance peripheral blood bone marrow, NGS peripheral blood bone marrow, Myeloid neoplasm peripheral blood bone marrow. ^2^ Detected clones. Pts, indicates patients; BM indicates bone marrow; PB, peripheral blood; AML, acute myeloid leukemia; MDS, myelodysplastic neoplasia; MDS/MPN, myelodysplastic neoplasia/myeloproliferative neoplasia; MPN, myeloproliferative neoplasia; NGS, next generation sequencing; ITD, internal tandem duplication; TKD, tyrosine kinase domain; NHL, non-Hodgkin lymphoma; MRD, minimal residual disease; ALL, acute lymphatic leukemia; TCR, T-cell receptor; IGH, immunoglobulin heavy chain; MPAL, mixed phenotype acute leukemia;

**Table 3 cancers-15-02305-t003:** NGS sensitivity and specificity.

		Peripheral blood		
		Positive	Negative	Total
**Bone marrow**	**Positive**	656	8	664
	**Negative**	1	1840	1841
	**Total**	657	1848	2505

Concordance = 99.6% (i.e., 2496/2505 × 100). Positive prediction value (PPV) = 99.8% (i.e., 656/657 × 100). Negative prediction value (NPV) = 99.6% (i.e., 1840/1848 × 100). Sensitivity = 98.8% (i.e., 656/664 × 100). Specificity = 99.9% (i.e., 1840/1841 × 100).

**Table 4 cancers-15-02305-t004:** Discordant samples (n = 9): mutations found in only one of the paired samples (i.e., either in the peripheral blood but not in the bone marrow, or vice versa).

ID	Sex	Age at Initial Diagnosis	Initial Diagnosis	BM Blasts, %	PB Blasts, %	WBC, G/L	Mutations Detected in BM, n	Mutations Detected in PB, n	Discordant Mutation	Pathway	VAF in BM, %	VAF in PB, %
1	f	55	AML	1	0	4.6	2	1	NPM1	Nucleolar multifunctional protein	0.6	Not found
2	f	84	AML	87	1	12.5	6	7	ASLX1	DNA methylation related	Not found	1.0
3	f	83	MDS	2.5	0	5.2	6	5	SETBP1	DNA replication	1.1	Not found
4	f	81	AML	19	2	1.7	5	4	NRAS	RAS pathway	1.3	Not found
5	f	79	MDS	3	not done	1.5	2	1	RB1	Tumor suppressor	1.9	Not found
6	f	75	MPN	2.5	2	3.6	4	3	SRSF2	Splicing factor	2.2	Not found
7	f	70	MDS	8	0	2.5	1	0	ASLX1	DNA methylation related	2.5	Not found
8	f	76	AML	40	0	3.1	9	8	RUNX1	Transcription factor	3.7	Not found
9	f	70	Waldenstrom’s disease	2.5	0	6.0	1	0	IDH1	DNA methylation related	9.1	Not found

ID indicates patient identification number; BM, bone marrow; PB, peripheral blood; WBC, white blood cell count; VAF, variant allele frequency; f, female; DNA, deoxyribonucleic acid; AML acute myeloid leukemia; MDS, myelodysplastic neoplasia; MPN, myeloproliferative neoplasia.

## Data Availability

The datasets supporting the conclusions of this article are included within the article and the Appendix A. Data sharing of patient level data collected for the study is not planned. However, we are open to research questions asked by other researchers and we are also open to data contributions by others. Participation requests or potential joint research proposals can be made at any timepoint to the corresponding author via email (dr.lisa.pleyer@gmail.com) and are subject to approval by the AGMT and its collaborators.

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
