# Peer review of "Myeloid NGS Analyses of Paired Samples from Bone Marrow and Peripheral Blood Yield Concordant Results: A Prospective Cohort Analysis of the AGMT Study Group"

_cancers, 2023, doi:10.3390/cancers15082305_

Round 1

Reviewer 1 Report

The authors compared NGS results from paired BM and PB samples collected from patients with myaloid malignancies. They found a strong concordance between BM and PB with only a few discordant cases, mainly positive cases in BM and negative in PB, mainly due to low VAF. Importantly the study is designed in such a way that only a few days pass (maximum 2) between BM and PB and they believe this might be the reason for worse results in other studies in which the samples are taken at different time points. The study is well conducted and well done from a methodological point of view. the problem they face is felt because a follow-up on peripheral blood would make the evaluation more rigorous and the compliance of doctors and patients would be better. 

Minor comments:

I'm not convinced by the statement that BM is only done in 50% of cases at diagnosis. (the references refer to works from 2015). This is probably more tru during follow-up.  Perhaps this also varies from country to country and from pathology to pathology, and has probably also changed over time.

For a correct classification of the disease the BM is currently recommended.  I would not try to convey the message that peripheral blood NGS assessment can replace BM but that it may be helpful where BM is not feasible (for example in myelofibrosis BM aspirate can fail as correctly indicated by the authors) or for more accurate monitoring. 

The authors cited " Guidelines still state that BM specimen are a pre-requisite for the above. However these painful and time consuming procedures are only performed in 50%. 

I sugget to change this sentence because we do not have data that NGS in PB can be a surrogate of BM aspirate for a correct diagnosis and prognosis of the disease . We know that cytogenetic provides relevant information for diagnosis and prognosis in MDS for example and in the majority of the cases cytogenetic analysis requires BM. 

Reviewer 2 Report

Jansko-Gadermeir and colleagues have compared NGS panel results from 240 consecutive, prospective, paired bone marrow/peripheral blood samples from 187 patients. The results showed a high concordance indicating that peripheral blood samples may be used routinely rather than bone marrow samples. This is a useful study that would benefit from some additional information and other changes.

Comments

1.      What guidelines specify that bone marrow have to be used for NGS panel analysis? Peripheral blood samples are very commonly used in many healthcare settings (although many people prefer BM samples if they are available and a detailed comparison of PB vs BM is a worthwhile exercise).

2.      P values should be given to a meaningful degree of accuracy, e.g. 2 significant figures

3.      It is unclear why the analysis has not focused on the 216 samples that were taken on the same day.

4.      Figure 1 and text: ‘translocations’ should be changes to ‘fusion genes’?

5.      Standard units should be used, e.g. 109/L, not G/L

6.      Most centres use a cut off of 5% vaf for calling mutations using standard myeloid panel analysis. It would be useful to perform the comparison of PB vs BM using a 5% cut off. And using a 5% cut off it would also be interesting to describe in more detail the clinical features of the cases who showed a 2 fold in vaf for any mutation.

Round 2

Reviewer 2 Report

The changes that have been made have improved the manuscript. Re my previous comments 1 and 2, please change the r values to 2 significant figures in the abstract, text and figures (not the P values as originally suggested). The second point refers to the simple summary: 'Guidelines still state that bone marrow specimen are a pre-requisite .....' is incorrect and should be changed.

Author Response

Thank you once again for your time, your suggestions and the opportunity to submit another revised version. The two minor changes have been addressed.

The Reviewers’ comments are in black, and our responses in cursive blue. The text in the manuscript has been updated in the tracking mode.

The changes that have been made have improved the manuscript. Re my previous comments 1 and 2, please change the r values to 2 significant figures in the abstract, text and figures (not the P values as originally suggested).

r-values in the abstract, figures and in the text as well as in the appendix figures have been changed to 2 positions after the decimal point in following lines:

  • Line 45
  • Lines 50-53
  • Table 1
  • Line 265
  • Line 312
  • Figure 3
  • Lines 325-326
  • Figure 4 A-D
  • Line 340-341
  • Figure 5 A and B
  • Lines 452-453
  • Lines 456-457
  • Appendix Fig 16-26

The second point refers to the simple summary: 'Guidelines still state that bone marrow specimen are a pre-requisite .....' is incorrect and should be changed.

This was replaced with:

Lines 24-33

In these diseases, the bone marrow blast percentage, and hence bone marrow specimen remain pre-requisite for (i) accurate diagnosis and disease classification (e.g. [1–3]), (ii) risk stratification, which typically also includes the bone marrow blast percentage as well as conventional cytogenetics and/or fluorescence in situ hybridization from bone marrow specimen (e.g. [4–6]), and (iii) response assessment/disease monitoring during treatment (e.g. [5–10]). In addition, most large-scale reports on molecular data in these diseases were generated from bone marrow specimen only, and hence cannot eo ipso be extrapolated to results obtained in the peripheral blood [e.g. [11–15]). Several groups, including ours, report that bone marrow evaluations, which can be painful and time-consuming are only performed in ~50% of patients during follow-up outside of clinical trials, indicating a clinical need for surrogate samples.